# Determinants of pregnancy-induced hypertension on maternal and foetal outcomes in Hossana town administration, Hadiya zone, Southern Ethiopia: Unmatched case-control study

**Getachew Ossabo Babore**[1]*, **Tsegaye Gebre Aregago**[1], **Tadesse Lelago Ermolo**[1], **Mangistu Handiso Nunemo**[2], **Teshome Tesfaye Habebo**[3]

1 Department of Nursing College of Medicine and Health Science, Wachemo University, Hossana, Ethiopia, 2 Department of Public Health College of Medicine and Health Science, Wachemo University, Hossana, Ethiopia, 3 Department of Health Management and Economics, School of Public Health, Tehran University of Medical Science, Tehran, Iran

* gossabo2004@gmail.com

**Data Availability Statement:** All the necessary data and materials are incorporated in the article and Supporting Information. Any further requests

## Abstract

### Background

Globally, 292,982 women die due to the complications of pregnancy and childbirth per year, out of those deaths 85% occurs in Sub Saharan Africa. In Ethiopia, pre-eclampsia accounts for 11% of direct maternal deaths.

### Objective

To determine maternal and foetal outcomes of pregnancy-induced hypertension among women who gave birth at health facilities in Hossana town administration.

### Methods

Institutional based unmatched case-control study was conducted among women, who gave birth at health facilities from May 20 to October 30, 2018. By using Epi-Info version 7; 207 sample size was estimated, for each case two controls were selected. Two health facilities were selected using a simple random sampling method. Sample sizes for each facility were allocated proportionally. All cleaned & coded data were entered into Epi-info version 3.5.1 and analysis was carried out using SPSS version 20. Multivariate analysis was performed to determine predictors of pregnancy-induced hypertension at a p-value of <0.05.

### Result

Women between 18 to 41 years old had participated in the study with the mean age of 26.00 (SD ±4.42), and 25.87(SD ±5.02) for cases and controls respectively. Out of participants 21 (30.4%) among cases and 21(15.2%) among controls had developed at least one complication following delivery. 12 (17.4%) and 8 (5.7%) foetal deaths were found in cases and

will be addressed by the corresponding author upon reasonable request.

**Funding:** The authors received no specific funding for this work from any source. all of us doing for academic qualification and put finger print for scientific society.

**Competing interests:** The authors declare that no competing interests.

controls groups respectively whereas 15.6% from cases and 3.6% from controls groups women gave birth to the foetus with intra-uterine growth retardation. Women gravidity AOR = 0.32 [95% CI (0.12 0.86)], Previous history of pregnancy-induced hypertension AOR = 22.50 [95% CI (14.95 16.52)] and educational status AOR = 0.32[95% CI (0.12, 0.85)] were identified as predictor of pregnancy-induced hypertension.

## Conclusion

Women with a previous history of pregnancy-induced hypertension had increased risk of developing pregnancy-induced hypertension, whilst $\geq$ 3 previous pregnancies and informal educational status decrease odds of developing pregnancy-induced hypertension.

## Introduction

Pregnancy and childbirth are natural processes, which comes up with multiple consequences. A hypertensive disorder is one of the pregnancy consequences which is a major alarming cause for maternal, perinatal morbidity and mortalities [1]. The term hypertension in pregnancy is commonly used to describe a wide spectrum of the patient who may have only mild elevations in blood pressure to severe organ dysfunction. Thus, it is accompanied by minor to major complications. Worldwide hypertensive disorder in pregnancy/HDP affects 5–22% and it is responsible for 5–10% of complications in all pregnancies [2–4].

Among four classes of HDP, preeclampsia remains a leading cause, which needs rigorous public intervention for better outcome of foetus and mother, and Preeclampsia affects up to 5–8% out of all pregnancies [5]. Also, preeclampsia is a unique form of hypertension during pregnancy which usually occurs after 20 weeks of gestation. If the early investigation and appropriate management are not undertaken for the women diagnosed with pre-eclampsia. It progress to a severe form called eclampsia, which end-up with maternal as well as foetal adverse outcomes like abruption placenta, acute renal failure/ARF, intravascular coagulation, intra-uterine growth retardation/IUGR, and stillbirth [6]. Therefore, the origin for eclampsia is pre-eclampsia (Eclampsia is the definition of Preeclampsia plus $\geq$ +2 proteinuria plus the occurrence of convulsion or coma) [7].

Studies suggested that either pre-existing pregnancy-induced hypertension/PIH or pregnancy changes could be responsible for the occurrence of pre-eclampsia. In a multicentre study approximately, 30% of hypertensive disorders of pregnancy were occurred due to chronic hypertension while 70% of the cases were diagnosed as gestational hypertension or pre-eclampsia [8]. Regardless of new-onset or pre-existing occurrences, the harmful effects of preeclampsia and eclampsia upraised from mother to child, family to the country and its severity is from trivial to life-threatening. Still, it has remained a significant public health threat in both developed and developing countries [9].

PIH denotes women's systolic blood pressure/SBP $\geq$ 140mmHg, and diastolic blood pressure/DBP $\geq$ 90mmHg on two or more consecutive measures without proteinuria after 20 weeks of gestation; pre-eclampsia is characterized as when pregnant women presented with SBP $\geq$ 140mmHg and DBP $\geq$ 90mmHg on two or more consecutive measures within 4 hours interval with the presence of proteinuria that occurs after 20 weeks of gestation whereas eclampsia denotes the occurrence of convulsion plus proteinuria +2 or more and sign and symptom of severe pre-eclampsia for the women who fulfil the definition of PIH [10–12].

Pre-eclampsia and eclampsia are the second direct cause for maternal death which accounts for 10 to 15% of maternal deaths worldwide [13]. The incidence of pre-eclampsia has significant variation in different parts of the continents. For instance, 4% in Africa, 3.8% in Europe, and 4.2% in the western Pacific region [14]. Moreover, the prevalence of pre-eclampsia throughout the country has vast variation, in Jima University specialized hospital, it was 51.8%, three southwest Ethiopia hospitals 6.3% [15], and in seven Tigray hospitals 50% [16].

Globally, 292,982 women died due to the complications of pregnancy and childbirth. Out of those deaths, 85% have occurred in Sub Saharan Africa/ SSA, yet the majority of those deaths occurred in low resource settings, and most of them could have been preventable [17, 18]. Furthermore, the highest share of maternal death has been reported in Africa as compared to other regions. Maternal death due to pregnancy-related causes is 1 in 4,000 in Europe and 1 in 16 in African countries [18, 19]. Likewise, The probability of a 15 year-old girl eventually dying from a maternal cause in Africa was as high as 1 in 37- as compared to 1 in 3400 in the European region [20].

According to the latest joint trend review study in maternal mortality conducted by United Nation Population Division/UNPD, World health organization/WHO and World Bank, the proportion of mothers dying per 100,000 live births has declined from 380 to 210 in 1990 to 2013 [21]. Besides, there was a slight reduction in maternal mortalities in the last three consecutive Ethiopian demographic health surveys/EDHs; MMR was 667, 665 and 412 per 100,000 life birth and all those deaths might have happened as a result of direct or indirect causes [22]. On the other hand, a trend review study from 1980 to 2012 in Ethiopia, on maternal death reported that as a result of hypertensive disorder of pregnancy/HDP, maternal death has increased from 4%-29%. In-addition to the death trend, the review pointed out the major direct obstetric complications (sepsis, haemorrhage, unsafe abortion, obstructed labour) including pre-eclampsia, accounts for 85% of maternal death. Whereas pre-eclampsia solely accounts for 11% of maternal death [23, 24]. Whilst pre-eclampsia and Eclampsia contribute to 53% of maternal and 62.7% of perinatal complications during pregnancy and birth [25].

HDP especially preeclampsia, in primigravida women is 2 times more risky than multigravida [26, 27]. Impacts of pre-eclampsia and eclampsia are disproportional in both developed and developing countries which are seven times higher in developing countries than in developed worldwide [28].

Impacts of Pre-eclampsia and eclampsia on maternal and foetal outcomes are enormous, which results in life-threatening events to death. For instance, it increases the risk of placenta abruption, postpartum haemorrhage/PPH and intra-uterine growth retardation. According to the WHO multicentre survey, the risk of perinatal death among women with preeclampsia and eclampsia increased 3 and 5 folds respectively, as compared to women with no preeclampsia or eclampsia [26, 29, 30]. Still, preeclampsia is one of the major causes of perinatal death in developing countries, accounts for 20–50% of deaths [31]. In Ethiopia, eclampsia accounts for 35.7% of maternal death [32, 33].

Studies were done abroad and our country revealed that pregnancy-induced hypertension has been associated with poor maternal and prenatal outcomes and loos of life [34]. Case control hospital-based study done in India reported that 10, 8, 3 and 2 complications of Haemolysis, Elevated Liver enzymes, and Low Platelet count/HELLP syndrome, PPH, Infection and ARF respectively [35]. On another case control study conducted by Guduri GB revealed that there were 18%, 2% PPH and 36%, 7% preterm complication, among cases and controls respectively [36].

Studies done in different regional hospitals, Ethiopia reported various proportions of maternal complications and deaths following delivery occurred as a result of pre-eclampsia/ eclampsia. The Eastern part of, Ethiopia finding revealed that 53% maternal and 62.7%

perinatal complication with a fatality rate of 11% [25], in Woliata Sodo University teaching Hospital 48.89% perinatal complication and 8.89% perinatal deaths [37]. In Zewuditu Memorial Hospital, among women had developed PIH, 131 (52.4%) of them developed complication whilst 31.1% of them experienced with HELLP syndrome [38].

Different studies noted that previous history of pregnancy-induced hypertension, age, and educational status were independent risk factors for the development of preeclampsia. In addition to these, occupation, gravidity, family history of hypertension, gestational diabetic Mellitus and residence had a significant statistical association with preeclampsia [39–42].

Yet, there have been different studies conducted to explore PIH in Ethiopia, but there was no study conducted on predictors of pregnancy-induced hypertension in the study area. Thus, to come up with effective public as well as clinical intervention approach and strong policy development direction, conducting root cause identification research is essential. Therefore, the main objective of this study is to determine maternal and foetal outcomes of pregnancy-induced hypertension among women who gave birth at Public health facility in Hossana town administration, Hadiya zone, Southern Ethiopia: unmatched case control [Fig 1].

## Methods and material

### Study area

The study was conducted in Hossan town administration, Hadiya Zone, South Nation Nationality and People Regional state/SNNPR, Ethiopia. Hadiya zone has ten woredas and two town administrations. According to the Hadiya Zone Finance and Economic Development office statistics, the total population of the zone was 2,486,242 of which 1,218,258 were men whereas 1,267,983 of them were female.

Hossana town is located in the northern part of SNNPR state. It is 232 KM far from the country's capital city to the south and 120 KM from the regional capital town. The town administration is classified into 3 subs administrative with a total of 8 kebeles. According to the Hossana town administrative office, the current (2018/2019) projection estimated total population was 104,053 whereas 50,986 males and 53, 067 of them were females. Among the total town population, women within the reproductive age group encompass 24,244 from them the estimated number of women who are eligible to be pregnant in the current physical year were 8,388 [43].

The town has one teaching hospital which has been serving more than 3, 548,800 million people from the entire Hadiya Zone and partial part of kembeta and Silte Zone. As well, the town has three health centres, one private surgical hospital, and more than 15 private clinics.

### Study design and period

Institutional based unmatched case control study was conducted in OB/GYN department of the selected public health facilities, from May 20/2018 to October 30/2018.

**Cases.** All pregnant women who were on follow up after 20 weeks gestational age and visit health facilities for delivery service and screened as of having pregnancy-induced hypertension registered in the OB/GYN departments of the respective facilities.

**Controls.** Pregnant women who have no PIH in the same period and the same health facilities and who came for delivery service after 20 weeks of gestational age.

### Source population

The source population of the study were all women, who have been on follow-up care unit and visit facilities for delivery service in Public Health facilities those resided in Hossana town administration.

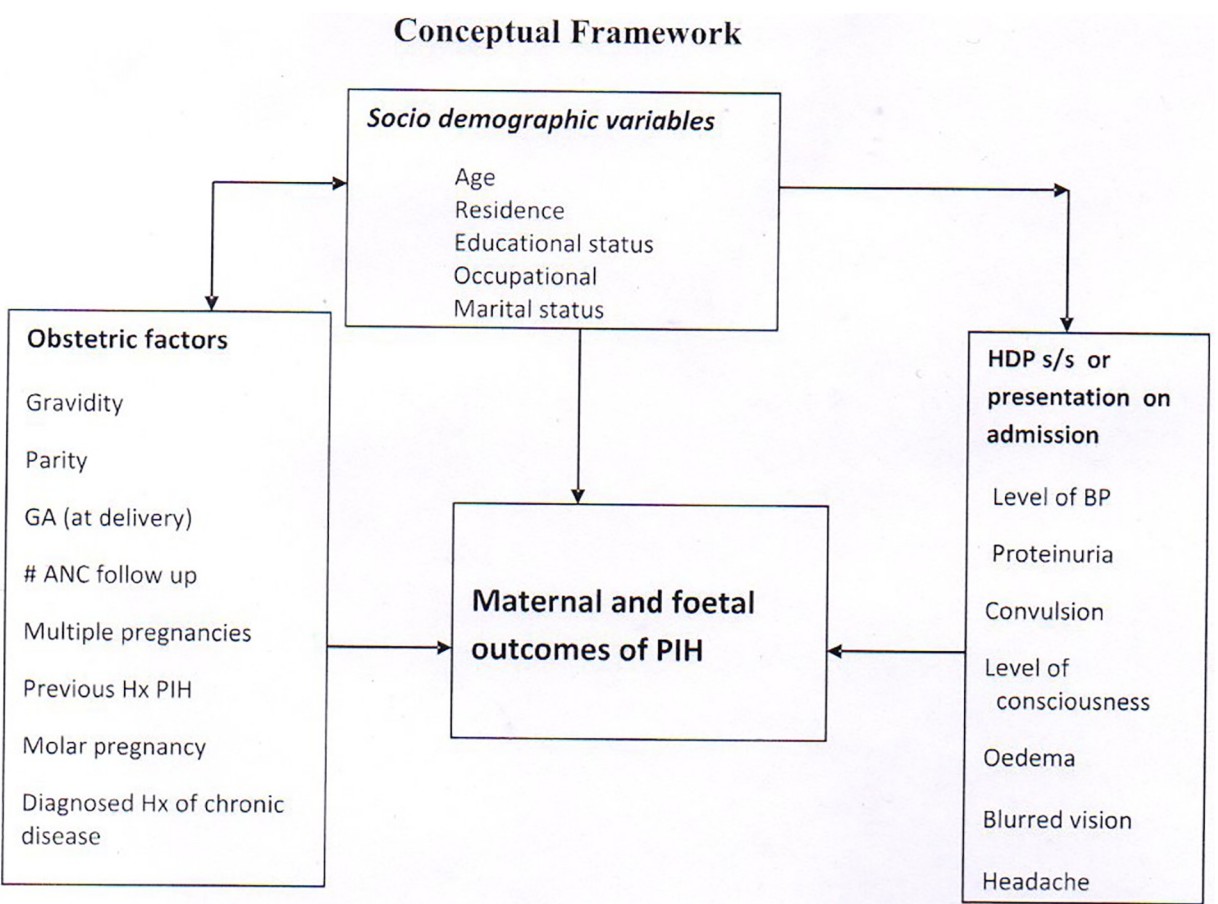

**Fig 1. Conceptual frame work of PIH and factors (source from literature review).** Legend: PIH: Pregnancy-induced hypertension, HDP: hypertensive disorder of pregnancy, Hx: history, ANC: antenatal care, GA: gestational age.

## Study population

All women who were selected using systematic sampling method applying population proportionate to sample size (PPS) from randomly selected public health facilities among women who had been on follow-up care unit and visit the health facility for delivery service whose gestational age above 20 weeks.

## Eligibility criteria

**Inclusion criteria.**   All pregnant women who were on flow up and visited selected public health facility for delivery service whose gestational age above 20 weeks were included. For women representing cases, they diagnosed having Pregnancy-induced hypertension as of her SBP $\geq$140mmHg and DBP $\geq$90 mmHg on two separate reading which measure within at least four hours apart, plus a dipstick reading +1 and above. For women represent Controls, women within the same health facilities who were attending delivery care was not diagnosed as having Pregnancy-induced hypertension.

**Exclusion criteria.**   Women who didn't indweller in respective town administrative sub towns and no longer stay at Hossan town administration for more than six months.

Women with a known diagnosis for Epilepsy and women who were not voluntary to give consent also excluded from the study.

## Simple size and sampling technique

**Sample size.** The sample size for this study was computed based on the comparison of proportion for case control study by using Epi-info version 7 for windows. According to a study conducted by Eskeziaw Agedew [39], by considering the factors gravidity and maternal age had an association with PIH. Being multigravida and age during current pregnancy between 25–30 years which have a significant association with pregnancy-induced hypertension with case to control ratio 1:2 and Odds ratio (OR) = 4 and using the following assumptions: power 80%, confidence level 95% (Table 1). The final sample size was taken from the women who were multigravida by adding 10% non-response rate. Thus, an estimated sample was employed for case 69 and 138 for controls yielding a total sample of 207.

## Sampling technique

To select a study unit two Public Health facilities were selected randomly among four facilities. Considering the two months report data from the health management information system/ HMIS office sample size allocated proportionally by using proportionate to population size for cases and controls. All newly registered pregnant women who were more than 20 gestational weeks suffered from pregnancy-induced hypertension were selected representing the cases. For each case, women who registered for ANC follow up and had given delivery whose gestational age ≥20 weeks, but hadn't experienced with PIH at the same time in the same facilities were taken as control. To select controls, a list of total women the MCH department registration book for those who have ANC follow up after 20 weeks of gestation age was considered as a sampling frame. The estimated sample size for this study (n) was divided by a total number of women (N) registered in randomly selected HFs during the last two months which yield proportionate (P). Then through multiplying proportionate value with two months sample, a proportional sample was allocated for each selected health facilities. Finally, by employing a systematic sampling method based on the $k^{th}$ value sampling unit was traced in respective facilities. The sampling procedure was presented (Fig 2).

## Data collection method, tools and procedures

Structured and pretested questionnaires which was prepared in English and translated to Amharic and then translated back into English again to assure consistency of tool, which developed from reviewing different literature was used in this study.

Data were collected by 4 BSc midwives, 2 BSc Nurses supervised and monitored during the data collection phase by using structured questionnaires whereas the principal investigator undertakes the overall coordination. Data were collected from women who gave birth in the OB/GYN department, for each case, two controls were interviewed on the same day and health facilities. Participant's medical charts were also reviewed to obtain biomedical laboratory data at the same time.

**Table 1. Sample size calculation for second specific objective for PIH effect on maternal and foetal health among women gave birth in public health facilities.**

| Variables | Expected frequency of control among exposed | Case | Control | OR | Total sample size |
|---|---|---|---|---|---|
| Multigravida | 10.1% | 63 | 126 | 4 | 207 |
| Age 25–30 | 38.4% | 16 | 155 | 4.59 | 171 |

**OR:** Odds ratio

# Sampling frame

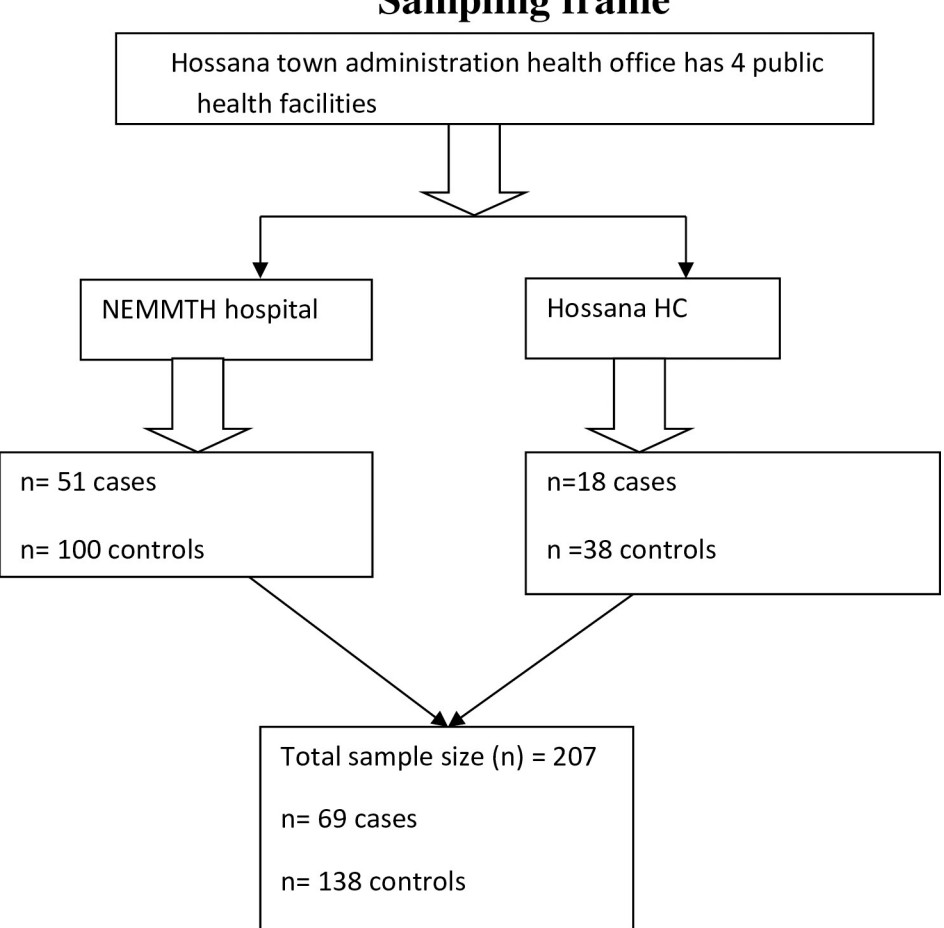

**Fig 2. Sampling procedure scheme of mother who gave birth in public health facility, Hossana town administrative, Southern Ethiopia, 2018.** HC: health centre, NEMMTH: Nigest Eleni Mohomod Memorial Teaching Hospital, n: proportional allocated sample size.

## Data quality control measures and management

**Training.**  Before the actual data collection date, data collectors & supervisors were trained concerning the overall issue of data collection format like, in time data collection following delivery, completeness, participant confidentiality and consistency.

**Pre-test.**  One week before the actual data collection date, research tool was tested on 13 women who gave birth at Doyogena primary hospital, validity checked then a lot amendment was undertaken.

Every day, the principal investigator and supervisors were checked data for completeness and incomplete questionnaires were discarded. Cross-checked and coded data were entered into Epi-info software version 3.5.1. For further analysis and data cleaning, it was exported to SPSS (statistical package for social science) version 20.

## Data analysis

To identify the proportion of the pregnancy induce hypertension impact on maternal and foetal outcome in-relation to outcome variable cross-tabulation frequencies were done. Cross

tabulation was also employed to test the relation of variables against with outcome variable. Bivariate logistic regression was conducted to select candidate variables for multivariate analysis at P-value < 0.05. Finally, to determine predictors of outcome variables multivariate analysis was employed.

Data was described and presented using cross-tabulation value to for descriptive findings and interpreted by looking at a variable that has an association with outcome variables on multivariate analysis with a 95% confidence interval for AOR.

## Study variables

**Dependent variable.**    Foeto-maternal outcomes of Pregnancy Induced Hypertension

**Independent variable.**    *Socio demographic variables.* Maternal age, residence, educational status, occupation and monthly income.

*Obstetrics variables.* Gravidity, number of parity, gestational age at delivery, number ANC follow up, number of babies (twin).

*Medical variables.* Previous history of pregnancy induced HTN, History of D/M, anaemia and renal disease.

*Hypertensive disorder sign and symptoms of pregnancies on admission.* Level of blood pressure during admission/presentation, vomiting, proteinuria, typical symptoms (oedema, blurred vision, headache and epigastric pain).

## Ethical consideration

Ethical approval was obtained from IRB of WCU, Official letter for zonal/woreda health department was written from University research and community service V/P office and a cooperation letter was written from respective woreda office managers to the randomly selected health facilities.

## Operational definitions of terms

**Hypertensive disorders of pregnancy.**    Includes chronic hypertensive and pregnancy induced hypertension, regardless of previous history of hypertensive disorder.

**Pregnancy-induced hypertension.**    Hypertension developed after 20 weeks of gestation where SBP $\geq$ 140mmHg and DBP $\geq$ 90mmHg, within two consecutive reading which measured 4–6 hours apart without proteinuria among mothers who were attended delivery unit at health facilities among previously normotensive women.

**Preeclampsia.**    The new onset of hypertension with SBP $\geq$ 140 and DBP $\geq$ 90 mmHg, within two reading which measured 4–6 hours interval with the presence of proteinuria with or without oedema which occurred after 20 weeks of gestation.

**Eclampsia.**    Mother with DBP greater than or equal to 110mmHg after 20 weeks of gestation and in-addition to the features of pre-eclampsia having one or more episode of convulsion or coma plus proteinuria [+]2 or more.

**Pregnancy outcome.**    Any women who had at least one prenatal as well as maternal unfavourable outcome after delivery

**Maternal complication.**    Any mothers had at least one complication among who had attended delivery at Hospital.

**Gestational age.**    The duration of gestation is measured from the first day of last menstrual period more than 20 weeks for this study.

**Foetal outcomes.**    Any diagnosed complication or death confirmed after delivery.

## Result

### Socio-demographic characteristics

Among a total of 207 participants, 69 cases and 139 controls women have participated in the study. Women from the age of 18–41 participated in the study while their mean age was 26.00 (SD ± 4.42), 25.87 (SD ± 5.02) for cases and controls respectively. Two women were participated in the study where their age was below legally eligible for marriage. The majority 97 (47.3%) participants' were house-hold wives, 9(4.3% of them were students and 69 (42.9%) had no formal education.

### Reproductive history and pre-existing medical illness during current pregnancy

Among women who had ANC visits three and above, 34 (52.3%) from cases and 72 (55.0%) from control were not developed PIH. Women with a high frequency of ANC follow-Up had a low proportion of becoming hypertensive on current pregnancy while among women who had no ANC follow-up, 4 from cases and 10 from controls groups had developed pregnancy-induced hypertension.

More than one-third of the study participants were primigravida among them 34 (31.5%) were not know their LNMP. However, 180 (86.9%) didn't attend the minimum expected ANC follow up, only 16 (7.7%) had four and above ANC follow up. Seven women among cases and 44 women from control groups gave birth by caesarean section, but the largest proportion 62 (89.8%) of women from cases gave birth via spontaneous vaginal delivery/SVD as compared with controls 94 (68.1%).

Among all interviewed women, 29 (14.0%) were experienced with pre-existing medical illness includes: diabetic Mellitus, anaemia and non-pregnancy induced hypertension 4.3%, 5.3% and 3.9% respectively. A high proportion of women from cases 14 (20.3%) had medical problems as compared to controls 15 (10.9%). Out of the total cases that participated in this study; twenty-four women had a previous history of PIH. On the occasion of health facility arrival, 120 (57.9%) women were admitted with pushing labour pain, but the remaining were came up with one or more features rather than labour pain (**Table 2**).

Even though the DBP range from 90 to 144 mmHg among cases the mean DBP, 104.13 (SD ± 9.20) was above the cut-off point of the normotensive women. Perinatal delivered from women with cases experienced with an average of 0.7 complications. On the occasion of reception for delivery service and follow-up care in addition to having high blood pressure, every woman from cases was admitted with at least of two suggestive clinical features for PIH whereas controlled had less than 1 clinical feature (Table 3).

### Maternal and fetal outcomes

A total of 69 women with cases participated in the study and showed a potential effect on maternal and perinatal health. Among the interviewed cases, 11 (15.9%) of them were developed eclampsia. On the occasion of the arrival to health facility among women who had developed eclampsia, only 3 (27.2%) of them were comatose on the occasion of the arrival of health facility. For all Women who developed PIH urine test was performed and a test result had shown a minimum +1 proteinuria for the dipstick test.

Among the total of interviewed cases, 21(30.4%) women have developed at least one complication following delivery. The majority of complications were 13 PPH and 7 disseminated intravascular coagulopathy/DIC. Moreover, PIH has a potential effect on maternal as well as perinatal outcomes; perinatal borne from women with PIH more likely to develop

**Table 2. Cross tabulation of socio-demographic and RH among women gave birth at public health facilities, in Hossana town administration, Southern Ethiopia, 2018.**

| Variable | Category | Participants | |
|---|---|---|---|
| | | Case n = 69 (%) | Control n = 138 (%) |
| Age of the women | | | |
| | 15–24 year | 25 (36.2) | 60 (43.5) |
| | 25–34 year | 40 (58.0) | 71 (51.4) |
| | 35–44 year | 4 (5.8) | 7 (5.1) |
| Educational status | No formal education | 26 (37.7) | 63 (45.7) |
| | Literate | 43 (62.3) | 75 (54.3) |
| Marital status | Single | 0 | 1 (0.7) |
| | Married | 67 (97.1) | 120 (87.0) |
| | Divorced | 2 (2.9) | 17 (12.3) |
| Gravidity | Primgravida | 20 (28.6) | 50 (71.4) |
| | Multigravida | 49 (35.8) | 88 (92.8) |
| ANC | Yes | 65 (94.2) | 128 (92.) |
| | No | 4 (5.8) | 10 (7.2) |
| Maternal complication | Yes | 21 (30.4) | 21 (15.2) |
| | No | 48 (69.6) | 117 (84.8) |
| Number ANC visit | 1–2 | 31 (47.7) | 59 (45) |
| | >3 and above | 34 (52.3) | 72 (55.0) |
| History of medical illness | Yes | 14 (20.3) | 15 (10.9) |
| | No | 55 (79.7) | 123 (89.1) |
| History of previous PIH | Yes | 24 (34.8) | 4 (2.9) |
| | No | 45 (65.2) | 134 (97.1) |

PIH: pregnancy-induced hypertension, ANC: antenatal care

complication than normotensive women. Out of 55, alive births among cases 32 (58.2%) had at least one complication, but out of total alive births, 32.9% (n = 68) perinatal hadn't any complication. From 20 perinatal deaths, 12 (17.4%) was reported from women who had developed PIH. Among 13 foetal IUGR, 8 of them were from cases as far as women diagnosed for PIH 3.7 times more risky to causes foetal IUGR than normotensive women (Table 4).

**Table 3. Mean score and proportion of selected items among women gave birth in Hossana town administration, southern Ethiopia, 2018.**

| | | Case | Control |
|---|---|---|---|
| Mean age of the respondents | | 26. 00 (SD ± 4.42) | 25.87 (SD ± 5.02) |
| Mean score of the SBP | | 157.32 (SD ± 18.89) mmHg | 126.02 (SD ± 13.75) mmHg |
| Mean score of the DBP | | 104.13 (SD ± 9.20) mmHg | 76.31 (SD ± 7.68) mmHg |
| Average number of suggestive CF for PIH on admission | | 2.5 (SD ± 1.14) | 0.22 (SD ± 0.59) |
| Number of maternal complication | | 0.66 (SD ± 0.74) | 0.60 (SD ±0.74) |
| Average number of Perinatal complication | | 0.65 (SD ± 0.95) | 0.54 (SD ± 0.83) |
| Parity | 0 | 13 | 97 |
| | 1 and above | 56 | 41 |
| Mode of delivery | SVD | 62 | 94 |
| | CS | 7 | 44 |

CF: clinical feature, CS: cesarean section, SD: Standard Deviation

**Table 4. Maternal and foetal outcome among women gave birth in Hossana town administration, southern Ethiopia, 2018.**

| Variable | Category | Maternal and fetal outcome among both group | | OR | 95% CI | P value |
|---|---|---|---|---|---|---|
| | | Case n = (%) | Control n = (%) | | | |
| Maternal complication | Yes | 21 (30.4) | 21 (15.2) | 2.83 | 1.30, 6.15 * | 0.01 |
| | No | 48 (69.6) | 117 (84.5) | | | |
| Perinatal complication | Yes | 32 (58.2) | 78 (63.4) | 2.23 | 0.87, 5.74 | 0.10 |
| | No | 23 (41.8) | 45 (36.6) | | | |
| HEELP syndrome | Yes | 4 (8.2) | 1 (1.2) | 0.14 | 0.02, 1.25 | 0.09 |
| | No | 45 (91.8) | 83 (99.8) | | | |
| PPH | Yes | 13 (27.7) | 13 (15.5) | 2.09 | 0.87, 4.99 | 0.10 |
| | No | 34 (72.3) | 71 (84.5) | | | |
| Neonatal death | Yes | 12 (17.4) | 8 (5.7) | 3.42 | 1.32, 8.82 * | 0.01 |
| | No | 57 (82.6) | 130 (94.2) | | | |
| IUGR | Yes | 8 (15.6) | 5 (3.6) | 3.71 | 1.16,11.88 * | 0.03 |
| | No | 53 (76.8) | 123 (89.3) | | | |
| Birth weight | Normal | 35 (63.6) | 91 (71.1) | 1.41 | 0.72, 2.74 | 0.32 |
| | LBW | 20 (36.4) | 37 (28.9) | | | |
| Gestational age at the delivery | Pre-term | 8 (11.6) | 6 (4.4) | 2.84 | 0.95, 8.55 | 0.06 |
| | Term | 61 (88.4) | 130 (95.6) | | | |

*statistically significant at 95% CI with P value < 0.05, OR: odds ratio, CI: confidence interval

## Pregnancy-induced hypertension and associated factors

Binary logistic regression with a confidence level of 95%, ($\alpha = 0.05$) was conducted and variables which have statistically significant at p-value < 0.05 were selected as candidate variable for the last model that determine predictors of pregnancy-induced hypertension among women gave birth at health facilities.

Finally, variables entered into the last model and multivariate analysis was performed. The Previous history of pregnancy-induced hypertension increased odds of developing pregnancy-induced hypertension by 22 folds, [95% CI (6.313, 80.204)], three and above previous pregnancies decreases odds of pregnancy-induced hypertension AOR = 0.32 [95% CI (0.12, 0.86)] and women who had no formal education, 68.4% [95% CI (0.12, 0.85)] less likely to develop PIH than women had primary and above educational status. Thus, the model was identified; gravidity, educational status, and previous history of pregnancy-induced hypertension were determinant factors for pregnancy-induced hypertension. Furthermore, Pregnancy-induced hypertension had an impact on inducing maternal complication, perinatal death and Intra-Uterine growth retardation/IUGR (Table 5).

## Discussion

This study revealed that 21(30.4%) among cases and 21(15.2%) women in control groups had developed at least one complication following delivery. The finding is supported by the study conducted in India 54% among case and 9% from controls developed maternal complications [36]. Also, a study done by Kapil Dev revealed that among cases 24% of women developed at least one maternal complication, but there was no maternal complication in controls [37]. A lower proportion of maternal complication in this study could be due to living style and women in the study area had less history of medical complications.

The commonest maternal complications in this study were postpartum haemorrhage/PPH 13 (18.8%), which is higher than the study done in India [35]. The lower proportion reported

**Table 5. Predictors of PIH among women gave birth at Hossana town administration, Southern Ethiopia, 2018.**

| Variable | Category | COR 95% CI | AOR 95% CI |
|---|---|---|---|
| Number of pregnancy/gravid | 1 times | 1 | 1 |
|  | 2 times | 2.07 (1.04, 4.13) | 1.09 (0.51, 2.34) |
|  | ≥3 times | 0.80 (0.37, 1.74) | 0.32 (0.12, 0.86)** |
| Educational status | College and above | 1 | 1 |
|  | Primary–high school | 0.48 (0.22, 1.04) | 0.54 (0.21, 1.39) |
|  | No formal education | 0.45 (0.21, 0.97) * | 0.32 (0.12, 0.85)** |
| Previous history of PIH | No | 1 | 1 |
|  | Yes | 17.87 (5.87, 54.27) | 22.50 (6.31, 80.20)** |

*statistically significant in bivariate analysis

** statistically significant in multivariate analysis, COR: crude odds ratio, AOR: adjusted odds ratio.

from elsewhere might be a better management approach and health care setups in those facilities were more intensive and organized as compared to our study area.

This study showed that perinatal complications were more prevalent among controls (63.4%) as compared to cases (58.2%). The finding is supported by the study done by Aleem Arshad, only 1 and 13 low birth weight reported from cases and control, respectively [44]. In this study, neonatal death was the second leading outcome of PIH, 17.4% from cases and 5.7% controls deaths were reported. Concerning perinatal complications, this study reported that 15.6% IUGR from cases and 3.6% from control which was lower than a study conducted in India; 29% and 71% IUGR were reported from women who had developed PIH and normotensive, respectively [36]. The possible reason for the low proportion could be socio-demographic factors and women in our study area affected by low superimposed medical problems.

Out of the cases, group preeclampsia accounted for 58(84.1%) whereas eclampsia comprises 11(15.9%). A study was done in Harare, Zimbabwe reported the proportion of pre-eclampsia and eclampsia were 1.7% & 0.3% respectively [9], but the proportion was lower than a study conducted by Selemawit, 121 women developed Pre-eclampsia and 17 of them Eclampsia [45]. The difference might be due to early identification and alerting women during ANC visits which decrease the possible occurrences of preeclampsia and eclampsia. Other study carried in three south-west hospitals, Ethiopia and tertiary care hospital of Visakhapatnam, India reported in the prevalence of pre-eclampsia and eclampsia were 7.9%, 3% and 16%, 36% respectively among cases [36, 46].

In this study gravidity has an association with PIH, women with gravidity 3 and above were 68% [95% CI (0.12, 0.86)], less likely to develop Pregnancy-induced hypertension as compared to their counterparts. This study is in-line with a study done in Darashe Special woreda [39] and Kombolicha, Ethiopia [41]. Whereas, the finding is in contrast with the study done in Addis Ababa, Ethiopia and Colombia; primigravida women were 2.7 times more likely to develop PIH than multigravida [24], and 36.9% of primigravida women among cases had developed PIH [47], the possible reason might be due to difference in assigning the reference group.

History of previous pregnancy-induced hypertension was significantly associated with PIH. In our study previous history of pregnancy-induced hypertension had 22 times increased odds of pregnancy-induced hypertension as compared to previous normotensive women. Out of the total interviewed women who had previous history of pregnancy-induced hypertension, 34.8% in cases and 2.9% in controls groups developed PIH in the current pregnancy. This finding is in-line with the studies done in Addis Ababa, Ethiopia and Karnataka, India. Women

with a previous history of pregnancy-induced hypertension were 4 times more likely to develop PIH during current conception, which reported 28.95% women among cases and 10.9% among control had developed PIH in Karnataka, India. Women with a previous history of PIH were 58 times odds of developing PIH, out of the total interviewed cases 60% and controls 2.50% of women developed PIH during current pregnancy [36]. However, the finding of this study contradicts with the study done in Jaipur, India [40].

In this study, multivariable analysis revealed that previous history of medical illness had no statistical association with PIH, but studies conducted in Tigray, Kombolicha in Ethiopia, and Southern India reported that women with diabetic Mellitus were 5.4, 11 and 5 times more likely to develop PIH respectively [16, 41, 48]. The possible reason for this discrepancy could be that the number of women with a pre-existing medical problem in this study was fewer than those studies conducted elsewhere.

In addition to the maternal complication, this study singled-out that perinatal complications such as low birth weight, IUGR, and pre-term were higher in the cases than controls. Also, the study demonstrated that there was no association between PIH and birth weight which is in contrast with the study done in Zimbabwe where women with PIH were 3 times more likely to have a baby with low birth weight [9]. However, this study is in-line with a study done by Eskzyiaw [39]. Major perinatal complications reported in this study were LBW (31.1%), IUGR (6.9%) and preterm (6.8%).

## Conclusion and recommendation

### Conclusion

Pregnancy-induced hypertension yet has been seen as a burning issue, which provokes adverse health impact on mothers and their babies. In this study, both maternal and perinatal outcomes were significantly different in both groups (cases & control). Women with PIH were at higher risk for a maternal and perinatal adverse outcome as compared to normotensive women. Women with a previous history of PIH had increased risk of developing PIH whilst women who had $\geq 3$ previous pregnancies and with informal education were less likely to develop pregnancy-induced hypertension.

### Recommendation

For women diagnosed with a previous history of pregnancy-induced hypertension, health care providers should have taken especial attention and focused care to tackle the adverse effect of PIH on their current conception. Furthermore, concerning governing bodies and partners engaged in maternal service should have facilitated basic setups like on-job training on early screening skills and managements, tax-free transportation. When gravidity increased women may not caution as like the first conception so that clinical expertise gave attention to alerting women regarding early warning sign and improve health service delivery strategies. Principal governing bodies and concern partners should have facilitated maternal waiting room/village for better health, good perinatal and maternal outcome.

## Supporting information

**S1 File.**
(DOCX)

**S2 File.**
(DOCX)

## Acknowledgments

First, of all, I would like to Praise and give thanks to God for his grace and blessings over this work. Secondly, I would like to thanks Wachamo University research and community service and vice President Office and College of Health science and medicine for overall coordination since preparation to finalizing the research project. Finally, my greatest gratitude extends to my NEMMRH MCH departments professionals and Hossana town administration health office for giving the baseline data for this study.

## Author Contributions

**Data curation:** Tadesse Lelago Ermolo.

**Writing – original draft:** Getachew Ossabo Babore, Mangistu Handiso Nunemo.

**Writing – review & editing:** Tsegaye Gebre Aregago, Teshome Tesfaye Habebo.

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
