## [Decision Letter · Decision Letter 0]

26 May 2020

PONE-D-20-09595

Determinants of pregnancy induced hypertension on maternal and foetal outcomes in
Hossana town administration, Hadiya zone, Southern Ethiopia: Unmatched case-control
study.

PLOS ONE

Dear Dr. Babore,

Thank you for submitting your manuscript to PLOS ONE. After careful consideration, we
feel that it has merit but does not fully meet PLOS ONE’s publication criteria as it
currently stands. Therefore, we invite you to submit a revised version of the
manuscript that addresses the points raised during the review process.

SPECIFIC ACADEMIC EDITOR COMMENTS: An expert reviewer in the field handled your
manuscript. Although interest was found in your study, several major comments arose
during review that require your attention. These comments include, but are not
limited to, the necessity to improve the readability of your manuscript and the
conclusions need to be better supported by the major results of this study.

Please submit your revised manuscript by Jul 10 2020 11:59PM. If you will need more
time than this to complete your revisions, please reply to this message or contact
the journal office at plosone@plos.org. When
you're ready to submit your revision, log on to https://www.editorialmanager.com/pone/ and select the 'Submissions
Needing Revision' folder to locate your manuscript file.

If you would like to make changes to your financial disclosure, please include your
updated statement in your cover letter. Guidelines for resubmitting your figure
files are available below the reviewer comments at the end of this letter.

We look forward to receiving your revised manuscript.

Kind regards,

Frank T. Spradley

Academic Editor

PLOS ONE

Journal Requirements:

4. Please upload a copy of Figures 1 and 2, to which you refer in your text. If the
figures are no longer to be included as part of the submission please remove all
reference to them within the text.

5. Please include a copy of Tables 1 to 5 which you refer to in your text.

Reviewers' comments:

Reviewer's Responses to Questions

**Comments to the Author**

1. Is the manuscript technically sound, and do the data support the conclusions?

Reviewer #1: Partly

2. Has the statistical analysis been performed
appropriately and rigorously? 

Reviewer #1: I Don't Know

3. Have the authors made all data underlying the
findings in their manuscript fully available?

Reviewer #1: Yes

4. Is the manuscript presented in an intelligible
fashion and written in standard English?

Reviewer #1: Yes

5. Review Comments to the Author

Reviewer #1: The paper entitled “Determinants of pregnancy induced hypertension on
maternal and foetal outcomes in Hossana town administration, Hadiya zone, Southern
Ethiopia: Unmatched case controlstudy,” is reporting on a case control study
undertaken in Hossana town administration, Hadiya zone, Southern Ethiopia, to
determine maternal and fetal outcomes of pregnancy induced hypertension.

It is commendable that the authors are researching how to improve care for women and
their babies during pregnancy in a low-income country with limited resources. I
congratulate them on conducting a case control study under these circumstances. The
finding that women with ≥3 pregnancies, previous history of pregnancy-induced
hypertension, and informal educational status were associated with pregnancy-induced
hypertension adds to knowledge in this area and may support early diagnosis and
improve management and pregnancy outcomes. These findings offer local evidence on
which to base future public health interventions in this population and may have
important implications for future research and practice. However, the paper requires
modifications before it is ready for publication.

Major:

1. The abstract conclusion and the paper conclusion are not consistent with the
reported results. The results report that having a previous history of
pregnancy-induced hypertension is associated with increased odds of
pregnancy-induced hypertension in the pregnant women in this study, and having ≥3
previous pregnancies, and no formal education, is associated with reduced odds of
pregnancy-induced hypertension. However, the abstract conclusion and paper
conclusion report that ‘women with multigravida, previous history of PIH and with
informal educational status were prone to develop pregnancy-induced hypertension.’
Please modify the abstract conclusion and paper conclusion so that they accurately
reflect the results.

2. The background and discussion contain considerable detail about other studies,
which is most interesting. However, the readability of the paper would be improved
if the background and discussion were restructured, so that the focus is on
information that is directly related to the objectives and findings of this study.
In addition, some comparisons made with other studies would benefit from rechecking
to ensure that the comparisons reported in this paper are accurate.

Minor:

3. The authors may wish to consider having the paper professionally edited for
English language to assist understanding.

4. The background reports ‘Pregnancy induced hypertension also known as
Preeclampsia…’ Please revise this definition, so that the reader understands that
while pregnancy-induced hypertension may lead to preeclampsia, they are not the same
condition.

5. When reporting the results of the multivariable analysis (which are odds ratios)
in the text, please consider describing the association of the assessed factors as
‘increased odds of pregnancy-induced hypertension’ or ‘decreased odds,’ rather than
‘likelihood’ or ‘prone to.’

6. Where percentages have been provided in the text, please also report the numbers
to support the readers understanding.

7. Please provide abbreviations in full the first time they are reported.

8. The authors may wish to consider combining some of the result tables and adding a
column for the univariable odds ratios for all factors in Table 2 to support
understanding. The addition of text below the tables to explain abbreviations in the
tables would also be helpful.

9. Please consider providing a copy of the data collection tool (the structured and
pretested questionnaire) as an attachment, as this would assist understanding of the
work that has been undertaken for this study and would be of considerable interest
to readers.

Thank you for the opportunity to review this most interesting and important case
control study that is researching the determinants of pregnancy-induced hypertension
in order to improve care for women and their babies in Ethiopia. I believe that
after some modifications, this paper will provide valuable local evidence on which
to base future public health interventions in this population.

6. PLOS authors have the option to publish the peer
review history of their article (what does this mean?). If published, this will
include your full peer review and any attached files.

If you choose “no”, your identity will remain anonymous but your review may still be
made public.

**Do you want your identity to be public for this peer review?** For
information about this choice, including consent withdrawal, please see our
Privacy Policy.

Reviewer #1: No

---

## [Author Response · Author response to Decision Letter 0]

22 Sep 2020

All authors of this manuscript read and took any effort to revise the whole
manuscript based on the comment given by peer review. For all forwarded minor and
major comments authors gave response as follows. 

1. We assure you that all components (manuscript, tables, figures, abstract and
abbreviations) of this article was prepared based on the PLOS ONE guide line. 

2. My ORCID is 0000-0002-7214-4025

3. Abstract in the manuscript and online submission in-terms of character and
contents were revised and identical. 

4. For all listed figure (figure_1 & Figure_2) in the main part of the manuscript
their copies uploaded separately. 

5. Copy of tables (table 1-5), which referred in the text were uploaded immediately
next to the each paragraph in the manuscript. 

Reviewer #1: The finding of this study revealed that women with previous history of
pregnancy induced hypertension more likely to develop pregnancy induced hypertension
on current conception whereas women with > 3 number of pregnancies and informal
education status were decreased odds of pregnancy induced hypertension. Thus,
research team/authors recommend that concern bodies should have facilitate basic
setups to make conducive environment (women waiting village, in-service training,
and tax free transportation) for better management of the cases where as health care
providers should have increased early diagnosis and on time intervention rather than
managing complication. Therefore, our paper conclusion part is modified as like
aforementioned. 

Major comment

1. Conclusion in abstract and paper regarding factors revised based on the finding of
the multivariate logistic regression. Variables description: women with previous
history of pregnancy induced hypertension increased odds of developing PIH in
current conception whereas number of pregnancies > 3 and women having informal
education status are 67.7%, 68.4% less likely to develop pregnancy induced
hypertension/PIH respectively, this implies they associated with reduced odds of
developing PIH. From the point of epidemiology, if the odd ratio/OR < 1 a factor
has protective effect of likelihood occurrence of event. Therefore, both number of
pregnancies > 3 and women having informal education status reduced odds of
developing pregnancy induced hypertension/PIH. 

2. In fact, background of this manuscript contains detail about others studies, but
for seek of readers better understanding and readability background of the
manuscript was revised based on the stated objectives and its finding. As you have
reviewed the back ground of the manuscript contained detail about the previously
studied articles, but the revised background contain about Pregnancy induced
hypertension of preeclampsia and eclampsia and maternal and foetal outcomes in
details. Moreover, discussion part also restructured through focusing on Objectives
(predictors of PIH and foeto-maternal outcomes) and for more understanding almost
all comparison was made with case control studies. We assure you that all comparison
made based on the reference which in-line with our study. For further accuracy and
consistent you can crosscheck according to the cited reference. 

Minor comment

3. The manuscript for better readability and understanding edited by English language
professionals who is PhD candidate in Addis Ababa University, Ethiopia. If you have
further enquires you can communicate through this address. Email negatuhabitamu@gmail.com cell phone +251912191000. 

4. Pregnancy induced hypertension and preeclampsia differ according to the
‘operational definition given in this manuscript. In-addition, Pregnancy induced
hypertension is a broad definition and according to the latest American college of
obstetrics and gynaecologist preeclampsia is one category of Pregnancy Induced
hypertension/PIH. In-fact, Preeclampsia may lead to ecalmpsia but not Pregnancy
induced hypertension lead to preeclampsia. A woman diagnosed with preeclampsia who
experienced with convulsion with or without coma can be diagnosed as eclampsia. 

5. Words ‘likelihood’, ‘prone’ used to describe finding of the multivariate analysis
are revised according the given comment. In main manuscript multivariable analysis
finding (Odds ratio) revised as previous history of pregnancy induced hypertension
is increases odds of developing PIH in current conception whereas informal
educational status decrease odds of PIH or less likely develop pregnancy induced
hypertension. 

6. For all reported percentages in the text cross-ponding ‘numbers’ also added. For
further confirmation you can check highlighted text under revised manuscript. 

7. Standard Abbreviation appears first in text fully described and appears in text
less than three times were excluded. 

8. As our understanding if we combine any one of the table; readers may be confused
because their descriptions and contents are different. For those abbreviations in
the table which are not described under abbreviation explanations in text are given
the tables. 

9. All structured and pretested questionnaires are uploaded as supporting
document.

to reviewer .docx
---

## [Decision Letter · Decision Letter 1]

14 Oct 2020

PONE-D-20-09595R1

Determinants of pregnancy induced hypertension on maternal and foetal outcomes in
Hossana town administration, Hadiya zone, Southern Ethiopia: Unmatched case-control
study.

PLOS ONE

Dear Dr. Babore,

Thank you for submitting your manuscript to PLOS ONE. After careful consideration, we
feel that it has merit but does not fully meet PLOS ONE’s publication criteria as it
currently stands. Therefore, we invite you to submit a revised version of the
manuscript that addresses the points raised during the review process.

There are still revisions that need to be addressed by the authors. You must respond
all of the reviewer's comments in your revised manuscript.

Please submit your revised manuscript by Nov 28 2020 11:59PM. If you will need more
time than this to complete your revisions, please reply to this message or contact
the journal office at plosone@plos.org. When
you're ready to submit your revision, log on to https://www.editorialmanager.com/pone/ and select the 'Submissions
Needing Revision' folder to locate your manuscript file.

If you would like to make changes to your financial disclosure, please include your
updated statement in your cover letter. Guidelines for resubmitting your figure
files are available below the reviewer comments at the end of this letter.

We look forward to receiving your revised manuscript.

Kind regards,

Frank T. Spradley

Academic Editor

PLOS ONE

Reviewers' comments:

Reviewer's Responses to Questions

**Comments to the Author**

1. If the authors have adequately addressed your comments raised in a previous round
of review and you feel that this manuscript is now acceptable for publication, you
may indicate that here to bypass the “Comments to the Author” section, enter your
conflict of interest statement in the “Confidential to Editor” section, and submit
your "Accept" recommendation.

Reviewer #1: (No Response)

2. Is the manuscript technically sound, and do the data
support the conclusions?

Reviewer #1: Partly

3. Has the statistical analysis been performed
appropriately and rigorously? 

Reviewer #1: I Don't Know

4. Have the authors made all data underlying the
findings in their manuscript fully available?

Reviewer #1: Yes

5. Is the manuscript presented in an intelligible
fashion and written in standard English?

Reviewer #1: No

6. Review Comments to the Author

Reviewer #1: The revised paper entitled “Determinants of pregnancy induced
hypertension on maternal and foetal outcomes in Hossana town administration, Hadiya
zone, Southern Ethiopia: Unmatched case control study,” is reporting on a case
control study undertaken in Hossana town administration, Hadiya zone, Southern
Ethiopia, to determine maternal and fetal outcomes of pregnancy induced
hypertension.

Thank you for the opportunity to review this interesting and important case control
study that is researching the determinants of pregnancy-induced hypertension in
order to improve care for women and their babies in Hadiya zone, Southern
Ethiopia.

The work by the authors to revise the manuscript based on previous feedback is
appreciated, however, further changes are required before the paper is ready for
publication. Following changes, this paper will provide valuable local evidence on
which to base future public health interventions in this population.

Major:

1. The authors have had the manuscript edited for readability and understanding for
English language from a professional who is PhD candidate in Addis Ababa University,
Ethiopia. However, it is essential that the authors have the paper edited again for
English language by a person who is skilled in this area. English spelling and
language improvements are necessary throughout every section of the paper, so that
the information is clear and understandable and the data is not misinterpreted by
the reader.

Minor:

To assist with the recommended changes to the paper, the authors may find previously
published papers in their reference list, on the same topic of hypertension in
Ethiopia, are a helpful guide.

Introduction:

• This section would benefit from being shorter and clearer. English spelling and
language improvements may assist with this.

• The information provided on hypertension, pre-eclampsia and eclampsia is important
as background to the paper, however, the way it is currently written is confusing to
the reader. Please revise so that the reader is clear about the definition of
pregnancy induced hypertension and how it relates to pre-eclampsia and eclampsia,
and maternal and fetal outcomes. Some papers in your reference list may be helpful
with this, e.g. Berhe, A.K., Kassa, G.M., Fekadu, G.A. et al. Prevalence of
hypertensive disorders of pregnancy in Ethiopia: a systemic review and
meta-analysis. BMC Pregnancy Childbirth 18, 34 (2018). https://doi.org/10.1186/s12884-018-1667-7.

Results:

• Please note that this reviewer is not an expert on statistical analysis, therefore
is unable to comment on the sample size and process of analysis.

• Table 4 and Table 5, please change the ORs, CIs, and p=values in the tables from 3
decimal places to 2 decimal places (e.g. Table 4, Maternal complication, please
change to OR 2.83, 95% C 1.30, 6.15, P 0.01).

• Table 5, “Number of pregnancy/gravid” requires “>3 times” to be changed to ≥ 3
times.

• While Table 5 shows that having ≥3 previous pregnancies is associated with reduced
odds of pregnancy-induced hypertension, the text reports that ‘Number of pregnancy
women with three and above pregnancies increased odds of pregnancy induced
hypertension with AOR=0.32 [95% CI (0.121, 0.864)] than two and below.” Please
change the text so that it matches the data in the table.

Discussion:

• It is recommended that the first paragraph of the discussion describes if the
objectives of the paper have been met.

• Comparisons made with findings from other studies are useful, but not always
understandable, and would benefit from English language improvements to improve
reader understanding.

Conclusion:

This conclusion is unclear. English language improvements would help to ensure that
the study findings are not misinterpreted by the reader.

7. PLOS authors have the option to publish the peer
review history of their article (what does this mean?). If published, this will
include your full peer review and any attached files.

If you choose “no”, your identity will remain anonymous but your review may still be
made public.

**Do you want your identity to be public for this peer review?** For
information about this choice, including consent withdrawal, please see our
Privacy Policy.

Reviewer #1: No

---

## [Author Response · Author response to Decision Letter 1]

27 Nov 2020

All authors of this manuscript read and took any effort to revise the whole
manuscript based on the comment given by the reviewer. All authors try to
incorporate the given comment in the entire manuscript. For all forwarded minor and
major comments authors gave response as follows. 

All authors rigorously tried to address all points answered by reviewer, which stated
from 1-7. Then major improvements were made from abstract to recommendation.

Reviewer #1: 

Major

In addition to previous edition made by English language professional who is PhD
candidate in Addis Ababa University, Ethiopia, in this round additional edition to
improve readability and understanding made by Dr. Feleke Doyore (PhD, health
promotion and communication) Who is lecturer in Wachemo University) and English
language professional Mr Yirga H. (BA, MA) currently, who is teaching in Bobicho
preparatory school, Hossan, Ethiopia. 

Minor 

Regarding Pregnancy Induced Hypertension, Pre-eclampis and eclampsia; to improve
readability and understanding for the reader, detail definition written in the
background part (from line number 85-94). In addition to this, for further
understanding detail definition is also described in the operational definition part
(line number 324-332). 

The relation between pregnancy induced hypertension with preeclampsia and eclampsia
is one is the inclusive the other; when a women diagnosed for eclampsia also fulfil
diagnosis criteria of preeclampsia as well women diagnosed for preeclampsia fulfil
diagnosis criteria of pregnancy induced hypertension. Therefore, eclampsis =
definition of Preeclampsia + occurrence of convulsion or coma + >+2 proteinuria
whereas preeclampsia = definition of pregnancy Induced hypertension + presence of
proteinuria in dipstick test with or without oedema. 

The effect of the pre-eclampsia and eclampsia on the maternal and foetal health
explained in detail in the background part (from line number 132-153), which reflect
the related outcomes of preeclampsia and eclampsia. 

Sample size calculation was performed by using statistical software EPI-Info V 7 for
window considering all statistical assumptions and others those needed to estimate
sample size. Thus, software sample size estimation is more recommendable and it
gives large sample size than manual calculation. 

All figures of the ORs, CIs and P values in the tables 4 and 5 are changed to the 2
decimal places from 3 decimal places. 

Table 5 finding and text report regarding having >3 previous pregnancies and its
odds ratio edited as ‘’ decrease odds of pregnancy induced hypertension.’’

Discussion 

Discussion based on the objective of the study revised rigorously. It is the fact
that, the first objective of this study is to measure the outcomes of the pregnancy
induced hypertension among cases and controls group. Hence, the outcomes of
pregnancy induced hypertension on maternal and foetal health explained in detail and
based on the multivariate analysis predictors of pregnancy induced hypertension
discussed one by one. 

Conclusion

To improve understanding and readability for the readers, major amendment undertaken
in conclusion part based on the objective of the study and finding of multivariate
analysis.

to reviewres.doc
---

## [Decision Letter · Decision Letter 2]

6 Jan 2021

PONE-D-20-09595R2

Determinants of pregnancy induced hypertension on maternal and foetal outcomes in
Hossana town administration, Hadiya zone, Southern Ethiopia: Unmatched case-control
study.

PLOS ONE

Dear Dr. Babore,

Thank you for submitting your manuscript to PLOS ONE. After careful consideration, we
feel that it has merit but does not fully meet PLOS ONE’s publication criteria as it
currently stands. Therefore, we invite you to submit a revised version of the
manuscript that addresses the points raised during the review process.

ACADEMIC EDITOR: The reviewer and I still have major issues with the readability of
this manuscript. The reviewer has kindly offered suggestions to improve this issue.
However, it is requested that the authors contact a copyeditor to help with
improvement and proof of English grammar and syntax. Failure to do so will prohibit
acceptance of this manuscript. Please provide a markup of changes within your
revised manuscript.

Please submit your revised manuscript by Feb 20 2021 11:59PM. If you will need more
time than this to complete your revisions, please reply to this message or contact
the journal office at plosone@plos.org. When
you're ready to submit your revision, log on to https://www.editorialmanager.com/pone/ and select the 'Submissions
Needing Revision' folder to locate your manuscript file.

If you would like to make changes to your financial disclosure, please include your
updated statement in your cover letter. Guidelines for resubmitting your figure
files are available below the reviewer comments at the end of this letter.

We look forward to receiving your revised manuscript.

Kind regards,

Frank T. Spradley

Academic Editor

PLOS ONE

Reviewers' comments:

Reviewer's Responses to Questions

**Comments to the Author**

1. If the authors have adequately addressed your comments raised in a previous round
of review and you feel that this manuscript is now acceptable for publication, you
may indicate that here to bypass the “Comments to the Author” section, enter your
conflict of interest statement in the “Confidential to Editor” section, and submit
your "Accept" recommendation.

Reviewer #1: (No Response)

2. Is the manuscript technically sound, and do the data
support the conclusions?

Reviewer #1: Partly

3. Has the statistical analysis been performed
appropriately and rigorously? 

Reviewer #1: I Don't Know

4. Have the authors made all data underlying the
findings in their manuscript fully available?

Reviewer #1: Yes

5. Is the manuscript presented in an intelligible
fashion and written in standard English?

Reviewer #1: No

6. Review Comments to the Author

Reviewer #1: The third draft of the revised paper entitled “Determinants of
pregnancy-induced hypertension on maternal and foetal outcomes in Hossana town
administration, Hadiya zone, Southern Ethiopia: Unmatched case control study,” is
reporting on a case control study undertaken in Hossana town administration, Hadiya
zone, Southern Ethiopia, to determine maternal and foetal outcomes of
pregnancy-induced hypertension.

The work by the authors to revise the manuscript based on previous feedback is
appreciated. However, despite the authors’ revisions, the paper is still not
presented in an intelligible fashion. To be clear, there are English language
corrections required in every section (abstract, introduction, methods and material,
result, discussion and conclusion), every sub-section, every paragraph, and every
sentence in this paper.

In addition, the paper is still too long, with unnecessary repetition, and many
sentences still do not make sense. The definition of pregnancy-induced hypertension
and how it is related to pre-eclampsia and eclampsia must be revised and
clarified.

Spelling corrections required throughout the paper include, but are not limited
to:

- Please correct the spelling of the word ‘sever’ to ‘severe’ (note that several
corrections are required).

- Please correct the spelling of the word ‘per-eclampsia’ and ‘preeclampsia’ to
‘pre-eclampsia’ (note that several corrections are required).

- Please correct the spelling of the abbreviation ‘HEELP’ to ‘HELLP’ (note that
several corrections are required).

- Please correct the spelling of the word ‘pregnancy induced hypertension’ and
‘pregnancy-induced hypertension (note that several corrections are required).

- Please correct the spelling of the word ‘case-control’ to ‘case control’ (note that
several corrections are required).

- Please correct the spelling of ‘dipstic’ to ‘dipstick’.

- Please correct the spelling of ‘Intra uterine growth retardation’ to ‘Intra uterine
growth restriction’ (note that several corrections are required, although following
the first use of IUGR abbreviation in the introduction, they can be abbreviated to
IUGR).

- Please remove unnecessary capital letters in the middle of sentences (note that
several corrections are required).

Abbreviation corrections required include, but are not limited to:

- Introduction, paragraph three, first sentence: This is the first time the words
‘pregnancy-induced hypertension’ appear in the main body of the paper. Please add
the abbreviation PIH here. That is, the sentence should say, ‘Studies suggested that
either pre-existing pregnancy-induced hypertension (PIH) or pregnancy changes could
be responsible for preeclampsia occurrence.’

- Introduction, paragraph seven, second sentence: This is the first time the
abbreviation ‘HDP’ appears in the main body of the paper. It should appear the first
time the words ‘hypertensive disorders of pregnancy’ appear in the main body of the
paper (which is in the first paragraph of the introduction).

- Introduction, paragraph eleven, second sentence: This is the first time the
abbreviation ‘HEELP’ (which must be corrected to HELLP) appears in the main body of
the paper. Please add the definition here. That is, the sentence should include that
HELLP syndrome includes Haemolysis, Elevated Liver enzymes, and Low Platelets.

In summary, there are English language corrections required in every section
(abstract, introduction, methods and material, result, discussion and conclusion),
every sub-section, every paragraph, and every sentence in this paper. Until these
English language corrections are made, the paper is shorter, and the sentences make
sense, I cannot recommend this paper for publication.

7. PLOS authors have the option to publish the peer
review history of their article (what does this mean?). If published, this will
include your full peer review and any attached files.

If you choose “no”, your identity will remain anonymous but your review may still be
made public.

**Do you want your identity to be public for this peer review?** For
information about this choice, including consent withdrawal, please see our
Privacy Policy.

Reviewer #1: No

---

## [Author Response · Author response to Decision Letter 2]

2 Apr 2021

Response to reviewers

Under the guidance of corresponding authors, all authors read for long period and
they brought the revised manuscript to the table and revision was made together. The
whole manuscript revised based on the comment given by the reviewer though using
copyedit and others language academicians. For all forwarded major and minor
comments authors gave response as follows. 

All authors rigorously tried to address all points answered by reviewer, which stated
from 1-6. Then major improvements were made from abstract to recommendation
according the points raised by reviewers. 

Reviewer #1: 

In addition to previous edition made by English language professional who is
instructor at Bobicho high school, Hossana, Ethiopia, Mr Yirga H. (BA, MA) in this
round additional edition in every section (abstract to Conclusion) to improve
readability and understanding made by Amanuel Tirkaso (MA, PhD candidate) Who is
lecturer at English department, college of social science, Wachemo University,
Hossan

In order to avoid unnecessary repetition, exhaustive reading and formatting few lines
were undertaken. To minimize paper length, total words in main body of the
manuscript (introduction to conclusion) reduced from 5674 to 5,222 words. 

Regarding Pregnancy Induced Hypertension, Pre-eclampis and eclampsia definition and
relation; to improve readability and understanding for the reader, detail definition
written in the background part (from line number 83-89). In addition to this, for
further understanding detail definition is also described in the operational
definition part (line number 289--298). 

The relation between pregnancy induced hypertension with preeclampsia and eclampsia
is one is the inclusive the other; a woman diagnosed for pre-eclampsia characterised
with elevated blood pressure (SBP > 140 & DBP > 90mmHg) without presence
of proteinuria which developed after 20 weeks of gestation. Preeclampsia: two
reading of SBP > 140 mmHg & DBP > 90mmHg 4-6 hours apart after twenty
weeks of gestation with proteinuria 2+ or more whereas eclampsia is women with signs
and symptoms of preeclampsia plus convulsion with or without coma. Oligohdrouia or
anuria is present. Therefore, eclampsis = definition of preeclampsia + occurrence of
convulsion or coma + >+2 proteinuria whereas preeclampsia = definition of
pregnancy-induced hypertension + presence of proteinuria in dipstick test with or
without oedema. 

All miss spelt words edited appropriately. Abbreviation correction in case of first
appearance revised through out of the main body of the manuscript. Regarding Intra
uterine growth retardation/IUGR the right word is retardation but not restriction. 

Discussion 

Discussion based on the stated objective of the study revised rigorously. More
focuses of the discussion done considering studies done using case control design.
Hence, the outcomes of pregnancy induced hypertension on maternal and foetal health
explained in detail and based on the multivariate analysis predictors of pregnancy
induced hypertension discussed one by one. 

Conclusion

Based on the objective of the study, conclusion focus adverse outcomes of PIH on
maternal and foetal health and multivariate analysis findings.

to reviewres.doc
---

## [Editor Report · Decision Letter 3]

12 Apr 2021

Determinants of pregnancy induced hypertension on maternal and foetal outcomes in
Hossana town administration, Hadiya zone, Southern Ethiopia: Unmatched case-control
study.

PONE-D-20-09595R3

Dear Dr. Babore,

We’re pleased to inform you that your manuscript has been judged scientifically
suitable for publication and will be formally accepted for publication once it meets
all outstanding technical requirements.

Kind regards,

Frank T. Spradley

Academic Editor

PLOS ONE

---

## [Editor Report · Acceptance letter]

20 Apr 2021

PONE-D-20-09595R3 

Determinants of pregnancy induced hypertension on maternal and foetal outcomes in
Hossana town administration, Hadiya zone, Southern Ethiopia: Unmatched case-control
study. 

Dear Dr. Babore:

I'm pleased to inform you that your manuscript has been deemed suitable for
publication in PLOS ONE. Congratulations! Your manuscript is now with our production
department. 

Kind regards, 

on behalf of

Dr. Frank T. Spradley 

Academic Editor

PLOS ONE